# Generalized Hartmann-Shack array of dielectric metalens sub-arrays for polarimetric beam profiling

Zhenyu Yang[1], Zhaokun Wang[1], Yuxi Wang[1], Xing Feng[1], Ming Zhao[1], Zhujun Wan[1], Liangqiu Zhu[1], Jun Liu[1], Yi Huang [1], Jinsong Xia[1] & Martin Wegener[2]

To define and characterize optical systems, obtaining the amplitude, phase, and polarization profile of optical beams is of utmost importance. Traditional polarimetry is well established to characterize the polarization state. Recently, metasurfaces have successfully been introduced as compact optical components. Here, we take the metasurface concept to the system level by realizing arrays of metalenses, allowing the determination of the polarization profile of an optical beam. We use silicon-based metalenses with a numerical aperture of 0.32 and a mean measured focusing efficiency in transmission mode of 28% at a wavelength of 1550 nm. Our system is extremely compact and allows for real-time beam diagnostics by inspecting the foci amplitudes. By further analyzing the foci displacements in the spirit of a Hartmann-Shack wavefront sensor, we can simultaneously detect phase-gradient profiles. As application examples, we diagnose the profiles of a radially polarized beam, an azimuthally polarized beam, and of a vortex beam.

[1] Wuhan National Laboratory for Optoelectronics, School of Optical and Electronic Information, Huazhong University of Science and Technology (HUST), 430074 Wuhan, Hubei, China. [2] Institute of Nanotechnology and Institute of Applied Physics, Karlsruhe Institute of Technology (KIT), 76021 Karlsruhe, Germany. These authors contributed equally: Zhenyu Yang, Zhaokun Wang, Yuxi Wang. Correspondence and requests for materials should be addressed to Z.Y. (email: zyang@hust.edu.cn) or to J.X. (email: jsxia@hust.edu.cn)

Amplitude, phase, polarization, and wavelength are basic parameters of any light wave. Of course, all of these parameters can be characterized experimentally by existing bulk optics. However, miniaturization often makes a big difference concerning usefulness. For example, established Hartmann–Shack wavefront sensors based on arrays of refractive microlenses, which focus the light onto a standard camera system, can simply be placed into a laser beam to analyze its phase and amplitude profile in real time[1]. Clearly, the same task can be performed by yet more compact metalenses. Metasurfaces and flat lenses based thereupon have recently attracted considerable attention. For example, compact polarimeters[2–4], polarization-sensitive elements[5–10], holograms[11–13], couplers[14,15], and metalenses[16–26] have been demonstrated. Metasurfaces can be based on metals[27–31] or dielectrics[32–37]. The former exhibit larger material contrast, the latter lower losses. Metalenses can be polarization independent[38–40] and broadband[41–44].

Importantly, metalenses can do more than just copying what refractive microlenses can do. Metalenses can, e.g., be designed to exhibit a tailored polarization dependence[45–50]. On this basis, we generalize the idea of a Hartmann–Shack lens array to not only measure phase profiles but simultaneously map polarization profiles as well. Here, we propose a generalized Hartmann–Shack array based on $2 \times 3$ sub-arrays of all-dielectric transmission-mode metalenses, and realize it experimentally. The six different metalenses in each sub-array allow to fully determine the Stokes parameters in each pixel of the array. To validate the concept of this generalized "meta-Hartmann–Shack" array, we use it to characterize a radially polarized beam, an azimuthally polarized beam, and a vortex beam.

## Results

**Principle of metalens array design.** Our system shown in Fig. 1 consists of two main parts: The metalens array and a standard camera, onto which the common focal plane of all metalenses is imaged. Each pixel of the metalens array is composed of one sub-array of six different metalenses (Supplementary Fig. 1). Each one of these metalenses focuses one particular polarization state (Supplementary Fig. 2). Six different polarizations are needed to fully and generally reconstruct the four Stokes parameters $s_0$, $s_1$, $s_2$, and $s_3$ We chose as the basis horizontal linear polarization ("x"), vertical linear polarization ("y"), diagonal linear polarization ("a"), 90 degrees rotated diagonal polarization ("b"), as well

as left-handed circular polarization ("l") and right-handed circular polarization ("r").

The unit elements of all metalenses shown in Fig. 2a are elliptical silicon pillars, with height $H = 340$ nm and major and minor axis lengths $D_x$ and $D_y$, respectively, placed on a layer of silicon dioxide. They are arranged onto a square lattice with lattice constant $a = 1500$ nm. To get an overview, we first consider a simple periodic lattice. Its intensity transmission and phase shift $\varphi$ versus $D_x$ and $Dy$, numerically computed using a finite-difference time-domain (FDTD) approach (Supplementary Note 1), are shown in Fig. 2b, c, respectively. Here we consider normal incidence and linear polarization of light along the x-direction.

A lens in the xy-plane made from these elliptical elements needs to satisfy the equation[18]

$$\varphi(x,y) = -\frac{2\pi}{\lambda}\left(\sqrt{x^2 + y^2 + f^2} - f\right) + \text{const.} \qquad (1)$$

Here, $\lambda$ is the free-space wavelength and $f$ is the metalens focal length. In this work, we choose $\lambda = 1550$ nm and $f = 30$ μm. The metalens arrays for "y", "a", and "b" polarization are simply rotated versions of that for the "x" polarization. For all of these linear incident polarizations, we obtain a theoretical (measured) focusing efficiency of 62% (30%). (We define the focusing efficiency by the ratio of the optical power in the focal spot and that impinging onto the metalens.) The design of the "l" and "r" metalenses is described below.

Figure 2d shows an optical image of a fabricated metalens array. The standard sample processing starts from a silicon-on-insulator (SOI) substrate (Supplementary Note 2 and Supplementary Fig. 3). Scanning electron micrographs of increasing magnification are shown in Fig. 2e–g, evidencing the high quality of the structures. Each metalens has a footprint of $(22.5\,\mu m)^2$. Together with the focal length $f$, this footprint leads to a numerical aperture of 0.32.

With the above metalenses alone, one cannot distinguish between left-handed and right-handed circularly polarized light. To eliminate this shortcoming, the "l" and "r" metalenses for circularly polarized light are designed by using the geometric phase shift also known as Pancharatnam-Berry phase shift[51–53]. Here, all silicon elliptical pillars have the same sizes $D_x$ and $D_y$. The phase variation $\varphi(x,y)$ of the metalens is exclusively achieved by the ellipse orientation angle $\theta_0$ (see Fig. 2a). Equations 2 and 3

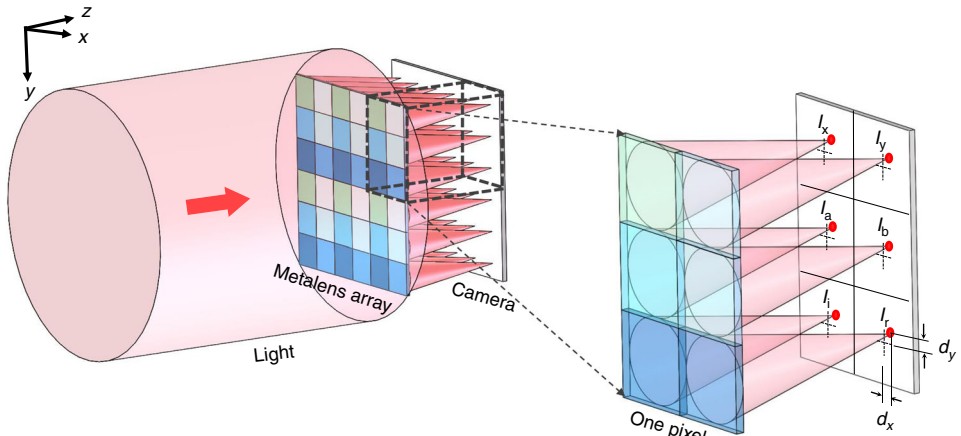

**Fig. 1** Scheme of the generalized Hartmann–Shack beam profiler. To measure not only phase gradients but also allow for complete polarimetric beam profiling, each pixel of the array shown on the left consists of six different polarization-sensitive metalenses. A camera records a magnification of the common plane of the metalens foci (magnification not depicted, see Supplementary Figs. 4 and 6). The dotted crosses on the camera plane in the right panel show the projections of the corresponding metalens centers

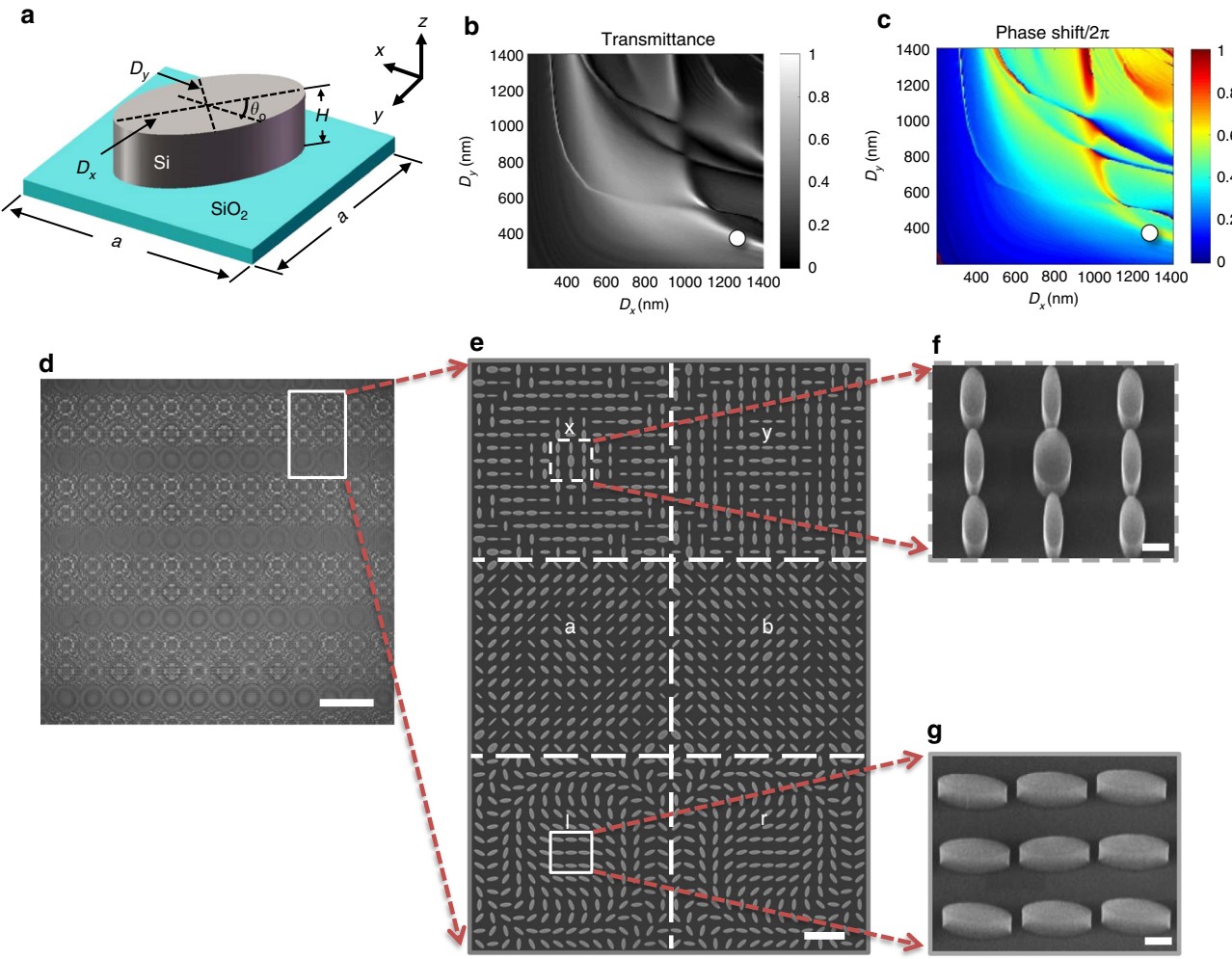

**Fig. 2** Design and manufactured metalens array. **a** Scheme of one unit element of a metalens. **b**, **c** Calculated intensity transmittance and phase shifts of a simple periodic array of these unit elements for linearly x-polarized incident light under normal incidence. The white circle at ($D_x = 1350$ nm, $D_y = 480$ nm) highlights the structural parameters used for the metalenses for circular polarization of light in our work. **d** Optical micrograph of a fabricated metalens array. Scale bar: 50 μm. The white rectangle indicates one pixel or one sub-array of the array. Each pixel is composed of six different metalenses. **e** Corresponding magnified electron micrograph. Scale bar: 5 μm. **f**, **g** Oblique-view and further magnified views onto selected parts of the metalenses, showing the individual high-quality silicon elliptical pillars (compare panel **a**). Scale bar: 500 nm

describe the complex transmission vectors for incident right-handed and left-handed circularly polarized light (see subscripts), respectively[19,52],

$$t_r = \frac{t_o + t_e}{2} \cdot \begin{pmatrix} 1 \\ -i \end{pmatrix} + \frac{t_o - t_e}{2} \cdot \exp(-i \cdot 2\theta_o) \cdot \begin{pmatrix} 1 \\ i \end{pmatrix} \quad (2)$$

$$t_l = \frac{t_o + t_e}{2} \cdot \begin{pmatrix} 1 \\ i \end{pmatrix} + \frac{t_o - t_e}{2} \cdot \exp(i \cdot 2\theta_o) \cdot \begin{pmatrix} 1 \\ -i \end{pmatrix} \quad (3)$$

Here $t_o$ and $t_e$ are the complex transmission coefficients for incident light polarized linearly along the major and minor axes of the ellipse, respectively. The first terms correspond to the same handedness as the incident light, the second terms to the opposite handedness. We let the first terms vanish by choosing $t_0 + t_0 = 0$ (see white circle at (1350 nm, 480 nm) in Fig. 2b, c). In this case, the incident wave is fully converted into the opposite circular handedness, which can be modulated by the geometric phase shift[49]. This additional geometric phase shift covers the required range of $0...2\pi$ if $\theta_0$ varies from 0 to $\pi$ (see Fig. 2a). In this manner, all phases needed according to the metalens Eq. 1 are obtained, both for left-handed and right-handed circularly

polarized light, respectively. For the circular incident polarizations, we obtain a theoretical (measured) focusing efficiency of 60% (26%).

For each pixel of the array, the Stokes parameters are connected to the focal intensities, $I$ (with corresponding subscripts), of the six types of metalenses in the sub-array of this pixel through the Eqs. 4–7[2,54,55]

$$s_0 = I_x + I_y \quad (4)$$

$$s_1 = I_x - I_y \quad (5)$$

$$s_2 = I_a - I_b \quad (6)$$

$$s_3 = I_r - I_l \quad (7)$$

For local linear polarization of light, the Stokes parameters can be reduced to the polarization angle $\theta_p$ given by

$$\theta_p = \frac{1}{2} \tan^{-1} \frac{I_a - I_b}{I_x - I_y}. \quad (8)$$

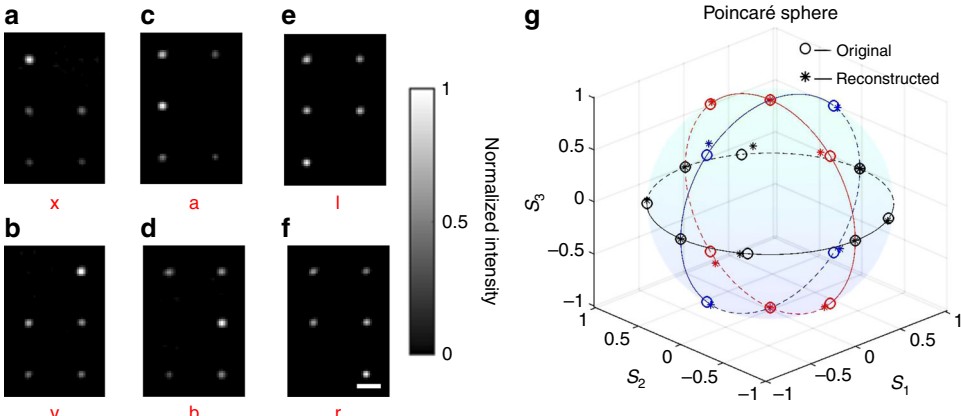

**Fig. 3** Experimental validation of polarimetry with only one pixel of the metalens array. **a–f** Measured images of the resulting focal spots for incident horizontal or vertical linear polarization ("x" and "y"), diagonal linear polarization ("a" and "b"), and circular polarization ("l" and "r"). Scale bars: 10 μm. **g** Poincaré sphere comparing the theoretical (small circles) and the experimentally reconstructed (stars) Stokes parameters $s_1$, $s_2$, and $s_3$. For each of the three different colors (red, blue, black), one Stokes parameter is zero

**Fig. 4** Profiling of vector beams by the metalens array. **a**, **b** Intensity distributions of focal spots for a radially polarized incident beam and an azimuthally polarized beam, respectively. The blue arrows qualitatively indicate the local polarization states. Scale bar: 50 μm. **c**, **d** Images of focal spots from the metalens array for the radially polarized and the azimuthally polarized beam, respectively. **e**, **f** Polarization profiles. The black arrows correspond to the measured local polarization vectors, the red arrows to the calculated ones. The dashed gray lines highlight the individual pixels of the metalens array

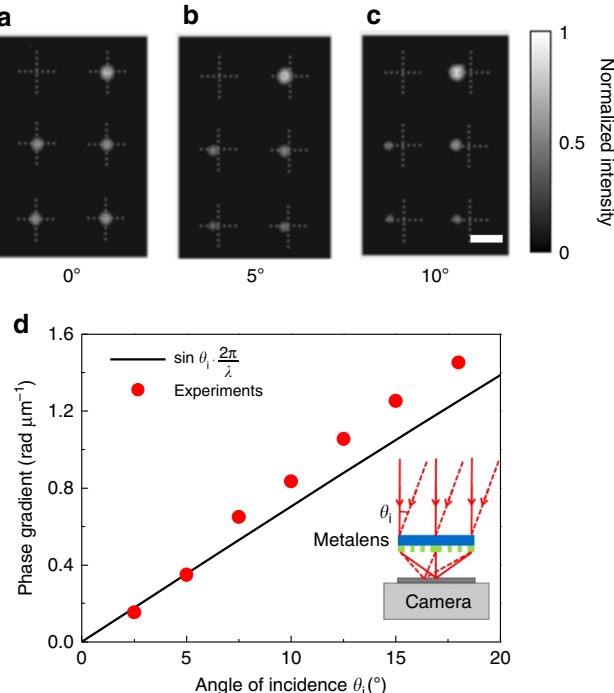

**Fig. 5** Experimental verification of wavefront detection by the metalens array. **a–c** Focal intensity distributions within one pixel of the metalens array for incident angles of 0°, 5°, and 10°, respectively. For clarity, dashed crosses are added to mark the central position of every metalens. Scale bar: 10 μm. **d** Phase gradient versus angle of incidence. Measurements are shown as red dots, theory by the black straight line

Finally, as in any ordinary Hartmann–Shack wavefront sensor, the local phase gradients along the $x$-direction and the $y$-direction are obtained via the expressions

$$\frac{\partial \phi}{\partial x} = \frac{2\pi}{\lambda} \cdot \frac{d_x}{\sqrt{f^2 + d_x^2}} \qquad (9)$$

$$\frac{\partial \phi}{\partial y} = \frac{2\pi}{\lambda} \cdot \frac{d_y}{\sqrt{f^2 + d_y^2}} \qquad (10)$$

where $d_x$ and $d_y$ are the displacements of the focal spot positions from the optical axis within the focal plane (see Fig. 1).

**State of polarization detection**. To validate the polarimetric beam profiling approach, we have performed experiments with 18 different incident polarizations (Supplementary Note 3, Supplementary Figs. 4 and 5). In this first set of polarimetric experiments, the polarization is constant throughout each incident beam profile. In order to ensure the accuracy of the measurement, we used "x", "y", "a", "b", "l", and "r" polarized light to calibrate the system. The details of the algorithm for the calibration are described in Supplementary Note 4. Figure 3a–f shows the raw data of the intensity distributions of one pixel of the metalens array for six selected different polarizations. The Stokes parameters of all 18 different polarizations are retrieved from the measurements and Eqs. 4–7. The input and the reconstructed Stokes parameters are compared on the Poincaré sphere in Fig. 3g. The average relative deviation between input and reconstruction is as small as 4.83%. Further experimental results are listed in Supplementary Table 1. Altogether, these results demonstrate that each pixel of the metalens array allows for

reliably determining the polarization state of light at the spatial position of this pixel.

In the second set of polarimetric experiments, we test the generalized Hartmann–Shack-array approach with two common beam profiles with non-constant polarization, namely with a radially polarized beam and with an azimuthally polarized beam. Figure 4a, b shows the beam intensity profiles for these incident two beams impinging directly onto the camera (i.e., no metalens array present). Inserting the metalens array in front of the camera, we obtain the raw arrays of focal spots shown in Fig. 4c, d, respectively. This particular metalens array is composed of 5 × 10 pixels. From these raw data, we derive the angle profile $\theta_p(x, y)$ according to Eq. 8. Clearly, we cannot derive polarimetric information near the center of the beams, where the intensity is close to zero. The results are depicted by the black arrows in Fig. 4e, f, respectively. The red arrows shown in these figures correspond to the theoretical values. The black and red arrows agree very well.

**Wavefront measurement**. The discussed polarimetric information goes beyond the usual Hartmann–Shack phase profiling. To validate this usual phase-profiling aspect as well, we have performed a first set of experiments with simple titled wave fronts with angles ranging from 0 to 18 degrees with respect to the surface normal. Resulting raw data are depicted in Fig. 5a–c. The displacements of the spot centers from the optical axis (indicated by the dashed crosses) are clearly visible. Through Eqs. 9 and 10, the phase-gradient profiles are retrieved. These values are compared with the calculated ones in Fig. 5d. It becomes clear that if the incident angle is less than 5°, the measurement agrees well with the theoretical value (relative error: ~1.05%); otherwise, the error increases (relative error of 18° incident angle: ~16.0%). This behavior is due to the fact that the metalens aberrations increase with increasing incidence angle.

In a second set of experiments, we further demonstrate the operation of the system for a more complex wavefront (Supplementary Fig. 6), namely for a vortex beam with a twisted wavefront and a topological charge of 3[56]. In this example, the metalens array consists of 7 × 4 pixels. Figure 6a shows the beam intensity profiles for the incident beam impinging directly onto the camera without passing through the metalens array. Figure 6b shows the raw data of metalens focal spots, and Fig. 6c the phase-gradient profile (see the arrows) derived from Eqs. 9 and 10. By integration, the phase profile is obtained (see false-color scale). We find a topological charge of the vortex beam of 3.25, which deviates by only about 8% from the input value of 3.

## Discussion

Since the data used in polarization and phase measurement are completely independent, the system can measure both polarization state and wavefront state at the same time, which is demonstrated in Supplementary Figs. 7 and 8. In addition, since the degree of polarization can be calculated via the Stokes parameters, it is clear that our system will also work for partially polarized beams. We demonstrate this aspect explicitly in Supplementary Figs. 9 and 10, and in the Supplementary Table 2.

In summary, we have demonstrated a dielectric metalens system not only allowing for measuring phase and phase-gradient profiles of optical beams (as a conventional Hartmann–Shack array), but also for measuring spatial polarization profiles at the same time. It is straightforward to mass fabricate these arrays of silicon-based sub-arrays, composed of six different metalenses, in a complementary metal oxide semiconductor (CMOS) compatible manner. Along these lines, it is also straightforward to substantially increase the numbers of pixels in the array, thereby

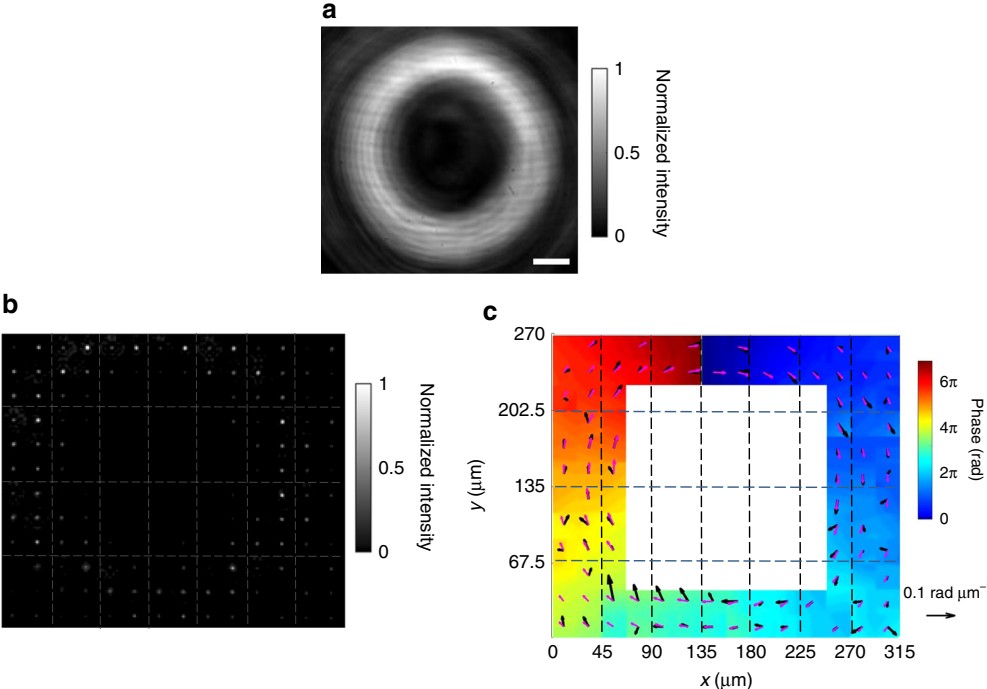

**Fig. 6** Profiling of a vortex beam by the metalens array. **a** Intensity profile for a vortex beam without the metalens array. Scale bar: 50 μm. **b** Raw data of measured focal spots of the metalens array. **c** Phase gradients (black arrows for the measured, pink arrows for the calculated) and wavefront (false-color scale) reconstructed on this basis. The dashed lines highlight the individual pixels of the 7 × 4 metalens array. The reference arrows has a length of 0.1 rad μm⁻¹

increasing the beam profiling resolution. The only other main component needed for complete polarimetric beam profiling is a standard camera connected to a computer. The system could even be integrated with a standard camera. In addition, the spatial resolution of the system can be further improved by reducing the diameter of the metalenses, the spacing of the unit cell, and the size of the sub-arrays. By replacing silicon with another dielectric, the operation principle can be transferred to other operation wavelengths. In our demonstration, we have used 1550 nm.

## Methods

**Metalens array design**. The characteristics of the unit cell in controlling the amplitude and the phase of the wavefront is analyzed using the FDTD method (Lumerical Inc. FDTD Solutions). In these calculations, the free-space wavelength of the incident light is set to 1550 nm and the linear polarization axis is parallel to the $x$-axis. We use periodic boundary conditions along the $x$-direction and the $y$-direction and perfectly matched layers (PMLs) along the $z$-direction. The geometrical parameters are given in the main text. Each pixel of the metalens array consists of six different metalenses, which are designed for "x", "y", "a", "b", "l", and "r" polarization. Through the data from Fig. 2b, c by applying Eqs. 1–3, the different metalenses are designed. The result is shown in Supplementary Fig. 1. It serves as the blueprint for the fabricated samples, examples of which are depicted in Fig. 2d–g.

**Fabrication process**. Prior to fabricating the metalens array, a double-polished SOI wafer is prepared by cleaning. Subsequently, a 430 nm thick ZEP520A electron-beam resist layer is spin-coated onto the wafer and subsequently baked for 3 min on a hot plate at 180 degrees Celsius to complete the curing of the photoresist. Next, to define the patterns, the sample is exposed by electron-beam lithography (EBL, Vistec: EBPG-5000+) at an acceleration voltage of 100 kV and at a beam current of 300 pA, followed by development in xylene and fixation by isopropanol. Afterwards, the sample with the patterned photoresist layers are etched by an inductively-coupled plasma system (Oxford Plasmalab: System100-ICP-180), using $C_4F_8/SF_6$ gas. The procedure is completed by removal of the patterned residual photoresist in an oxygen plasma (Diener electronic: PICO plasma stripper).

## Data availability

The data that support the findings of this study are available from the authors on reasonable request, see author contributions for specific data sets.

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

## Acknowledgements

This work was supported by the National Natural Science Foundation of China (NSFC, grant no. 61475058, grant no. 61335002, and grant no. 11574102), by the National High Technology Research and Development Program of China (grant no. 2015AA016904), and by the Fundamental Research Funds for the Central Universities (HUST, grant no. 2017KFYXJJ025). M.W. acknowledges support by the Helmholtz program "Science and Technology of Nanosystems" (STN) and by the "Karlsruhe School of Optics & Photonics" (KSOP). We thank the Center of Micro-Fabrication and Characterization (CMFC) of Wuhan National Laboratory for Optoelectronics (WNLO) at Huazhong University of Science and Technology (HUST) for the device fabrication support. We thank J. Jiang for his help in the fabrication, Y. Wang, C. J. Ke, P. Fei, and H. B. Yu for their help with the measurements.

## Author contributions

Z.Y.Y. had the original idea, conceived the study, and supervised the work; Z.K.W. performed the simulations and measurements with the help from X.F., M.Z., Z.J.W., and Y.H.; Y.X.W. fabricated the samples and analyzed them with the help of L.Q.Z. and J.L.; J. S.X. and M.W. supervised the fabrication and manuscript writing. All authors discussed the results. Z.Y.Y. wrote a first draft of the manuscript, which was then refined by contributions from all authors.

## Additional information

**Competing interests:** The authors declare no competing interests.

