## [Peer Review File · Nature Communications]

This manuscript has been previously reviewed at another journal that is not operating a transparent peer review scheme. This document only contains reviewer comments and rebuttal letters for versions considered at Nature Communications. Mentions of prior referee reports have been redacted.

Reviewers' Comments:

Reviewer #1:

Remarks to the Author:

The authors revised the manuscript and responded to the reviewers' comments. Regarding the response to Reviewer 1 comment 2, I have the following reply:

The authors wrote:

1. The system of "C" is able to measure the intensity, polarization, and phase at the same time, while "A" and "B" only discuss the measurement of intensity and polarization.
2. The design idea of "C" is based on the Hartmann-Shack wavefront sensing system. So in the design of "C", the projection of one metalens covers a number of pixels (16*16) on the CCD. "A" and "B", however, are based on the division of focal plane polarization cameras (DoFP-PCs), in which the projection of one metalens covers only one pixel on the CCD. Therefore, the systems in "A" and "B" can only detect the polarization state, but not phase.

My reply:

Yes, C provides more functionality at the cost of lower resolution compared to A and B, as any Hartmann-Shack wavefront sensing system does. There is no unambiguity about this. But "A" and "B" could provide the same functionality with a small change in design (i.e. use more pixels per metasurface lens/splitter).

The authors wrote:

3. The reviewer states that the efficiency of "B" is higher than that of "C". We believe that this comparison is not entirely appropriate: 1) "B" does not provide a specific definition of efficiency. 2) In "B", the wavelength is 850 nm, in which silicon will suffer great losses. However, in "C", the wavelength is 1550 nm, in which silicon has very low losses. Therefore, generally speaking, the transmittance of "C" should be greater than that of "B". 3) Furthermore, the efficiency is not a decisive parameter for our system anyway.

My reply:

"B" uses phase masks that simultaneously split polarization and phase, so both theoretically and in practice is more efficient than "C" which uses polarization filtering and thus has an upper limit of 50% efficiency. Regarding the material, "B" uses amorphous silicon which has negligible losses at 850nm, but as the authors say it is not a main point.

The authors wrote:

4. The unit elements in "C" are elliptical silicon pillars, while the unit elements in "B" are rectangular silicon pillars. In terms of fabrication, the rectangular structure is harder to handle than the elliptical structure. Therefore, comparing the SEM images, it is clear that the fabrication deviations in "B" are significantly larger than that of "C".

My reply:

This is a very specific comment that depends on how the device is made. Actually, rectangles are easier to handle in an e-beam writer than ellipses.

Regarding the rest of the reply, the authors responded satisfactorily to my comments.

Overall, my opinion is that this work is too incremental for Nature Communications. While there is some technical novelty in this paper, the Hartmann-Shack wavefront sensing system has been known for a long time, and there is nothing new about the implementation of the dielectric metasurfaces. I think this is good technical work that belongs to an OSA journal like Optics Letters or Optics Express.

Reviewer #2:

Remarks to the Author:

I had reviewed the first version of this manuscript submitted Nature Photonics and my assessment of the work was already fairly positive, but I had raised three issues. The authors have now answered to two of them in a satisfactory way (numbers 2 and 3), although they have limited all the changes to the supplementary information file and have not even mentioned these issues in the main paper (at least I did not find any comment). Moreover, in their rebuttal letter they have answered to my comments concerning issue 1, but they did not introduce any significant change to the paper. I think it would be better to include the requested discussion in the paper itself, not just in the answers to the reviewers. Perhaps, important issues that have been raised by several reviewers (some of my comments coincide with those of other reviewers) could at least be briefly mentioned in the main paper, even if most related material is put in the supplementary information. Overall, I am favorable to recommending this paper for publication in Nature Communications, but would suggest that in the final revision of the manuscript the authors adjust their main article in order to take into account the most important comments of the reviewers.

Reviewer #3:

Remarks to the Author:

I think that the authors have made a good attempt to answer all the questions and to improve the manuscript. I also agree that the phase and polarisation measurement with the same device is nice. As for the prior work, well, I guess this work has some advantages and further, I feel that they arrived for review at similar times.

Reviewer 1

Comment 1: “C” provides more functionality at the cost of lower resolution compared to A and B, as any Hartmann-Shack wavefront sensing system does. There is no unambiguity about this. But “A” and “B” could provide the same functionality with a small change in design (i.e. use more pixels per metasurface lens/splitter).

Our response: We thank the reviewers for agreeing that “C” has more functions. It is true that this multifunctionality comes at the cost of reduced spatial resolution. However, we respectfully emphasize that the design presented in our manuscript is an early proof-of-principle demonstration. We believe that through careful optimization of design and fabrication, the spatial resolution can be significantly improved in the future. Therefore, we have added an approach for improving the spatial resolution to the Discussion section.

Comment 2: “B” uses phase masks that simultaneously split polarization and phase, so both theoretically and in practice is more efficient than “C” which uses polarization filtering and thus has an upper limit of 50% efficiency. Regarding the material, “B” uses amorphous silicon which has negligible losses at 850nm, but as the authors say it is not a main point.

Our response: Efficiency of such system is only one aspect of its performance. The most important feature of the Generalized Hartmann-Shack system in our work is that it can simultaneously measure the amplitude, polarization and phase information of the light beam. Following upon our response to comment 1, the efficiency of our system can most likely be optimized and improved further in the future.

Comment 3: This is a very specific comment that depends on how the device is made. Actually, rectangles are easier to handle in an e-beam writer than ellipses.

Our response: In the actual processing, the fabrication of ellipses and rectangles have different merits and demerits, respectively. The electron-beam writer works in a rectangular style, which is advantageous for the making of rectangular patterns. However, perfect rectangular corners are difficult to fabricate due to the well-known proximity effect. It is indeed more difficult to generate pattern files for the elliptical structures. However, it is easier to reduce the fabrication errors for elliptical structures through compensation techniques. When comparing the SEM images, it is clear that the fabrication deviations in “B” are significantly larger than that of “C”. Furthermore, comparing the fabrication processes of “B” and “C”, the latter is simpler, hence advantageous. It is simpler because no hard mask is needed in the fabrication processes of “C”.

In summary, our work has a number of substantial differences and advantages with respect to previously published work. On this basis, we expect that our work will attract attention.

Reviewer 2

Comment 1: I had reviewed the first version of this manuscript submitted Nature Photonics and my assessment of the work was already fairly positive, but I had raised three issues. The authors have now answered to two of them in a satisfactory way (numbers 2 and 3), although they have limited all the changes to the supplementary information file and have not even mentioned these issues in the main paper (at least I did not find any comment). Moreover, in their rebuttal letter they have answered to my comments concerning issue 1, but they did not introduce any significant change to the paper. I think it would be better to include the requested discussion in the paper itself, not just in the answers to the reviewers. Perhaps, important issues that have been raised by several reviewers (some of my comments coincide with those of other reviewers) could at least be briefly mentioned in the main paper, even if most related material is put in the supplementary information. Overall, I am favorable to recommending this paper for publication in Nature Communications, but would suggest that in the final revision of the manuscript the authors adjust their main article in order to take into account the most important comments of the reviewers.

Our response: Thank you very much for this positive assessment of our work. According to your specific advices, we have revised the manuscript further.

1. The algorithm for the reconstruction of the Stokes parameters, which is discussed in the Supplementary Note 4, is mentioned briefly in the section State of polarization detection of the manuscript.
2. The reviewers asks whether the system has the ability to detect partially polarized light and whether it can simultaneously detect polarization and phase. The detailed answers to these two questions have been added to the Supplementary Note 3. Furthermore, we have also added a brief discussion on these two questions in the Discussion section of the manuscript.
3. Finally, some aspects on improving the spatial resolution of the system have been added in the Discussion section of the manuscript.

Reviewer 3

Comment 1: I think that the authors have made a good attempt to answer all the questions and to improve the manuscript. I also agree that the phase and polarisation measurement with the same device is nice. As for the prior work, well, I guess this work has some advantages and further, I feel that they arrived for review at similar times.

Our response: Thank you very much for this very positive overall assessment of our work.